# Fatigue Analysis of Long-Span Steel Truss Arched Bridge Part II: Fatigue Life Assessment of Suspenders Subjected to Dynamic Overloaded Moving Vehicles

**Peng Liu [1], Hongping Lu [1], Yixuan Chen [1], Jian Zhao [2], Luming An [2], Yuanqing Wang [3] and Jianping Liu [1,*]**

[1] School of Architecture and Civil Engineering, Shenyang University of Technology, Shenyang 110870, China; pliu@sut.edu.cn (P.L.); lhp19971008@163.com (H.L.); cyx980201@163.com (Y.C.)
[2] China Railway Bridge Engineering Group Co., Ltd., Beijing 100039, China; zhaojianll@126.com (J.Z.); anluming@stdu.edu.cn (L.A.)
[3] Department of Civil Engineering, Tsinghua University, Beijing 100190, China; wang-yq@mail.tsinghua.edu.cn
[*] Correspondence: liujianping024@sut.edu.cn

**Abstract:** In a half-through steel arched bridge, the suspenders are the critical load transfer component that transmits the deck system and traffic load to the arch rib. These suspenders are subjected to traffic and environmental vibrations and are prone to fatigue failure, especially for overloaded moving vehicles. This paper aims to study the impact of moving vehicles' overloaded rate on the fatigue performance of suspenders in a long-span three steel truss arch bridge. Based on the Mingzhu Bay steel arch bridge, a 3D finite element bridge model was first established and seven types of moving fatigue vehicle models were considered. Then the stress amplitude and dynamic response of the suspenders on the middle steel truss arch were studied under a standard, 25%, and 50% overloaded moving vehicles load. Following that, the Miner fatigue cumulative damage theory was employed to evaluate the fatigue life of the suspenders. The results show that the short suspenders in the middle steel truss arch have the shortest fatigue life but can still meet the design requirements under the standard load. However, the fatigue life of the suspenders decreases by 20% and 30% when the overloading rate reaches 25% and 50%. The fatigue life cannot meet the design requirement when the overload rate is 50%.

**Keywords:** suspender; steel arch bridge; fatigue life; overloaded rate; finite element

## 1. Introduction

The half-through steel truss arch bridge has been widely used in modern bridges due to its advantage of having excellent load capacity and a simple construction method [1]. The suspenders of the steel truss arch bridge are the key load-bearing component that transmit the deck system load, moving vehicle load, and wind load to the upper arch ribs. These suspenders are subjected to repeated cyclic stresses with a varying magnitude as a result of the dynamic vehicle loading during their lifecycle [2]. The short suspender of the arch bridge has a high amount of stiffness and is subject to high vibration frequency. The anchorage end of the arch bridge is in a frequent state of bending and shear stress and is vulnerable to rain and other corrosion effects [3]. For the dynamic load of the overloading vehicles in particular, the dynamic amplification effect can accelerate the fatigue damage as a result of the large stress ranges induced [4]. The accumulated fatigue damage can initialize the crack and may cause the bridge's failure. However, the fatigue design may underestimate the effect of overloading vehicles on the fatigue life of steel bridges [5]. For the fatigue life analysis of metal structures, Carpinteri et al. [6] reformulated frequency-domain critical-plane criteria to improve the accuracy of the fatigue life estimation of smooth metal structural members under multiaxial random loads. Maercek et al. [7] provided methods for the determination of energy-based characteristics of the fatigue life of materials and a new

approach for the determination of strain energy density parameters. Pejkowski et al. [8] studied small cracks on the surface of fatigue specimens under multiaxial loading and found that there was a correlation between the observed damage mechanism and the quality of fatigue life prediction.

Many efforts have been made to investigate the effect of dynamic vehicle loading on the fatigue life of suspenders. Zhang et al. [9,10] established the finite element model of suspenders, and the fatigue reliability of a long-span bridge was analyzed under the combined dynamic load of vehicles and the wind load. The results showed that wind load significantly increased the stress on the suspender, but had little impact on the fatigue damage of the suspender. The dynamic effect of vehicles on long-span bridges is little and the impact of vehicle speed and road roughness can be ignored. Yang et al. [11] analyzed the impact of five-axis vehicles on simply supported and continuous beam bridges; the parametric studies were conducted and an equation for continuous beams subjected to moving vehicle loads was proposed. The results showed that the deflection, bending moment, and shear force at the midpoint of simply supported beams are linearly proportional to the velocity parameters. Based on the field monitoring data, Kwon et al. [12,13] used the equivalent probability density function to evaluate the fatigue reliability of steel bridges and proposed a method to predict the fatigue reliability evaluation of bridges in terms of the probability of equivalent stress amplitude distribution. To improve fatigue life estimation, the bilinear *S–N* method was integrated into a probabilistic model to analyze the uncertainties related to the fatigue deterioration process, and a probabilistic method based on bilinear stress life was proposed. Raju et al. [14] proposed a fatigue evaluation and design method to assess the remaining life of existing bridges; the results showed that the proposed method can evaluate the safety index more accurately. In addition, fatigue design methods have been included in the AASHTO specification.

Due to the randomness and complexity of moving vehicle loads, many scholars have established numerical vehicle models to simplify moving vehicle loads, and the fatigue performance of bridges has been investigated. Mohammadi et al. [5,15] proposed a bridge rated model considering the influence of overloading fatigue damage. The model was applied to five bridges, and it was shown that the fatigue damage caused by overloading is significant for old bridges, Furthermore, frequent overloading accelerates fatigue damage and shortens the fatigue life of bridges. E. J. Obrient et al. [16,17] proposed a semi-parametric fitting program that can simulate the traffic load effect. A micro-simulation algorithm was proposed to predict more accurately the applied traffic load on the long-span bridge when two vehicles were mixed between lanes in a traffic jam. Schilling et al. [18–21] investigated the design value of the number of stress cycles caused by vehicles passing through the span of various types of steel-frame highway bridges and a numerical method was proposed. The results showed that vibration response and vehicle spacing have little impact on the design value of cyclic stress under standard conditions. Wang et al. [22,23] established a mathematical model for counting typical vehicles, compared the static effect of heavy vehicles with that of standard designed vehicles, and studied the influence of the road roughness correlation on dynamic influencing factors. The results showed that the tandem axle load of heavy vehicles exceeded the limit in the guidelines. Zhao et al. [24] conducted a statistical analysis of the maximum bending moment and shear force of simply supported, 2-span, and 3-span continuous beam bridges; the top 5% of heavy vehicles in each vehicle category were selected and the results showed that a 5-axle, short, single vehicle produces greater shear forces in bridge girders than standard Permit truck models, and the 5-axle vehicle was proposed for the standard Permit vehicle and bridge ratings. Zhang et.al. [25,26] established the three-dimensional vehicle model and 3D dynamic bridge model, and the effect of vehicle speed and road roughness conditions on existing bridges was investigated. A fatigue reliability assessment framework for existing bridges in their service life considering the random effects of vehicle speed and road roughness conditions was proposed. This method can be used to modify the fatigue damage of multiple stress

ranges with different amplitudes into one equivalent stress range cycle and shows more accuracy in a lognormal distribution to describe the modified equivalent stress range.

However, many studies mainly focused on investigating how the load affects the fatigue behavior of in-service bridges, and little attention has been paid to considering the effect of overloaded moving vehicles on the fatigue life of steel arch bridges. This paper aims to study the impact of moving vehicles' overloaded rate on the fatigue performance of suspenders in a long-span three steel truss arch bridge. Based on the Mingzhu Bay steel arch bridge, a 3D finite element bridge model was established and seven types of moving vehicle models were considered. The stress amplitude and dynamic response of the middle steel truss arch suspenders were studied under standard, 25%, and 50% overloaded moving vehicles. In addition, the Miner fatigue cumulative damage theory was employed to evaluate the fatigue life of suspenders subjected to standard and overloaded moving vehicles. The flow chart of this paper is shown in Figure 1.

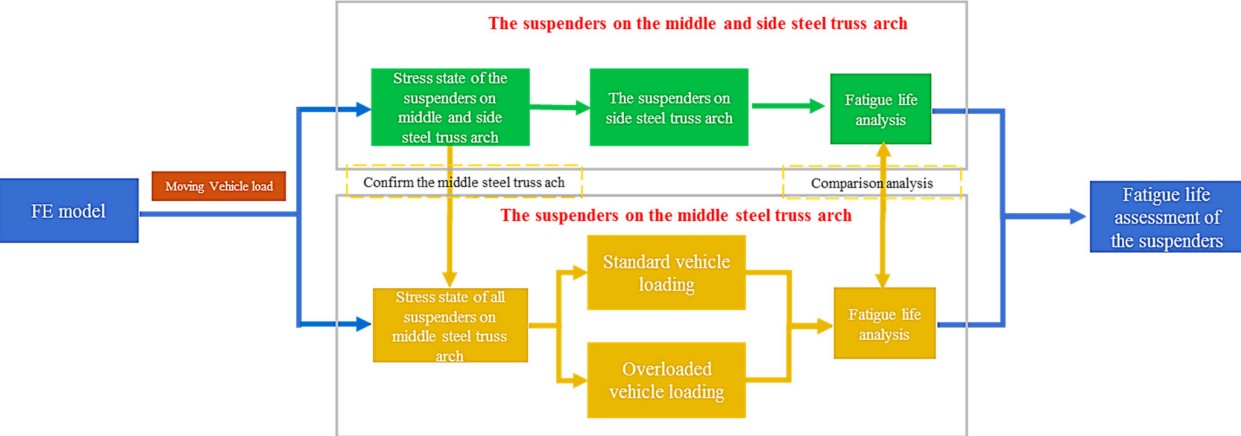

**Figure 1.** Flow chart of analysis.

## 2. Project Summary

The main bridge of Mingzhu Bay is (96 + 164 + 436 + 164 + 96 + 60 m) a six-span continuous three-steel truss arch bridge with a total length of 1016 m. Double orthotropic steel decks are used. The upper deck is a two-way eight-lane highway and there are sidewalks are on both sides. The total width of the deck is 43.2 m [24,27]. Both sides of the lower deck are reserved for the walking lanes. Figure 2 shows the bridge elevation and the schematic diagram of the side span and secondary side span of the main bridge.

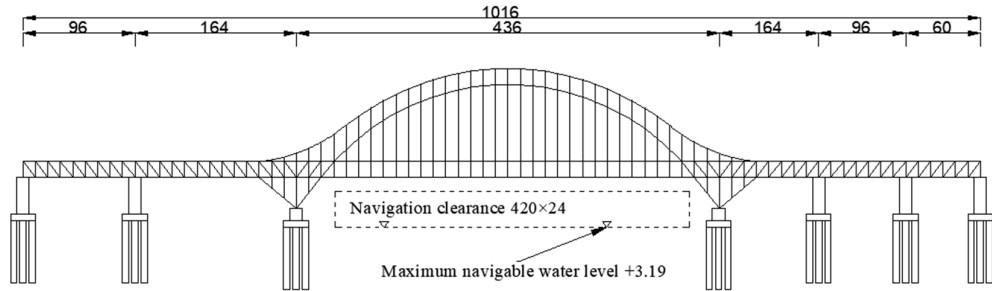

**Figure 2.** Elevation of Guangzhou Mingzhu Bay bridge (Unit: m).

## 3. Finite Element Analysis

### 3.1. FE Model of the Bridge

The 3D finite element bridge model was established by Midas/Civil [28], as shown in Figure 3. The whole model has 6676 nodes and 10,768 elements in total, which consist of 81 truss elements, 8067 beam elements, and 2620 shell elements.

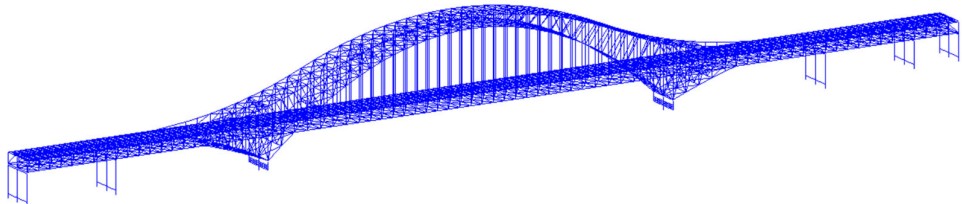

**Figure 3.** FE model of Guangzhou Mingzhu Bay bridge.

### 3.2. Suspenders' Parameters

Truss elements were used to simulate the suspender of the steel truss arch bridge. Since the steel truss arch bridge is symmetrical, the half-span suspender was selected for analysis. For the transverse direction of the bridge, the short suspenders of the middle truss arch and the side truss arch were selected for the comparative analysis of fatigue life, as shown in Figure 4a, while the numbers of the suspenders are shown in Figure 4b. Figure 5 illustrates the geometry of the suspender.

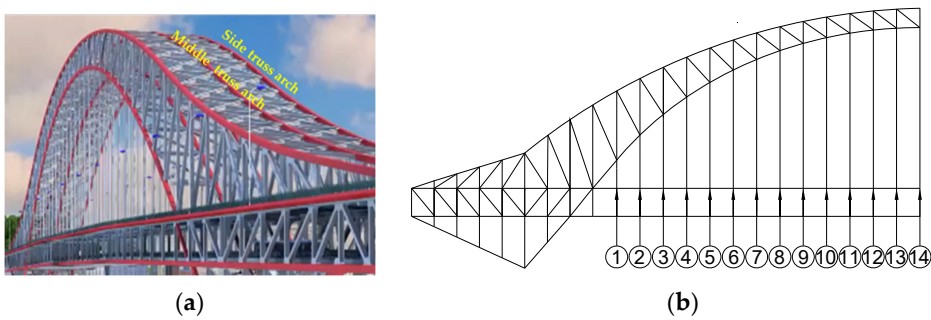

(**a**)                                                                 (**b**)

**Figure 4.** The steel truss arch and suspender number. (**a**) Steel truss arch; (**b**) the suspender numbers.

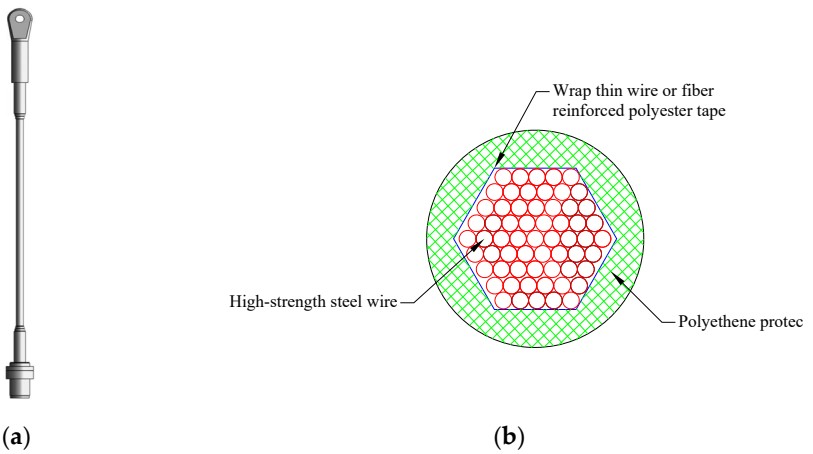

(**a**)                                                                 (**b**)

**Figure 5.** The geometry of the suspender. (**a**) The schematic of suspenders; (**b**) cross-section.

### 3.3. Chemical Composition and Mechanical Properties of Suspender Cable

The chemical composition and mechanical properties of suspenders are listed in Tables 1 and 2.

**Table 1.** Chemical composition of suspenders [29].

| Chemical Composition | C | Si | Mn | Cr | S | Cu |
|---|---|---|---|---|---|---|
| Mass fraction (%) | 0.85~0.90 | 0.12~0.32 | 0.60~0.90 | 0.10~0.25 | ≤0.025 | ≤0.10 |

**Table 2.** The mechanical properties of suspenders [30].

| Nominal Diameter (mm) | Tensile Strength (MPa) | Elongation | Modulus of Elasticity (MPa) |
|---|---|---|---|
| $7.0 \pm 0.07$ | 1670 | $\geq 4.0$ | $(2.0 \pm 0.1) \times 10^5$ |

## 4. Movie Vehicle Load and Fatigue Life Assessment Method

### 4.1. Moving Vehicle Load

The dynamic load of the vehicle is regarded as a spring-mass system with a uniform speed, while the design speed of the bridge is 80 km/h as simulated in the FE model. The vehicle load is considered as the harmonic force of uniform motion on the bridge deck. The vehicle is simplified to a uniformly moving biaxial mass. Based on the traffic volume statistics of Guangzhou Mingzhu Bay Bridge [31], the daily traffic flow can be predicted. According to the axle number and axle load, vehicles are divided into M1, M2, M3, M4, M5, M6, and M7 fatigue vehicle models. The vehicle load statistics and traffic flow statistics are shown in Table 3.

**Table 3.** Fatigue traffic model and traffic flow statistics.

| Vehicle Model Type | Vehicle Model (Axle Weight, kN, Axle Spacing, mm) | Total Weight (kN) | Daily Traffic Flow | The Proportion of Total Traffic |
|---|---|---|---|---|
| M1 |  | 135 | 1543 | 14.78 |
| M2 |  | 222 | 223 | 2.14 |
| M3 |  | 286 | 73 | 0.70 |
| M4 |  | 319 | 460 | 4.14 |
| M5 |  | 389 | 13 | 0.13 |
| M6 |  | 387 | 135 | 1.29 |
| M7 |  | 468 | 1453 | 13.92 |

### 4.2. Fatigue Life Assessment Method

Miner's linear cumulative fatigue damage theory [32] was adopted to evaluate the fatigue life of the suspenders. The fatigue damage of the suspender could be obtained from the *S-N* curve [33]. The fatigue life equation is listed in Equation (1):

$$\lg N = 15.1 + 13.5 \cdot \left( \frac{1}{\sigma_a} - \frac{\sigma_m}{\sigma_a - \sigma_b} \right) \tag{1}$$

where N is the number of cycles, $\sigma_a$, $\sigma_b$ are the tensile strength of the suspenders and the mean stress of the suspenders, respectively, and $\sigma_m$ is the mean stress value of $\sigma_a$ and $\sigma_b$.

## 5. Results and Discussion

In order to investigate the stress state of the suspenders under a moving vehicle load, the unit moving vehicle load was applied to the bridge deck in the FE model. The vehicle started at 10 s and passed through the long-span arch bridge deck at 30 s. The stress amplitude of each suspender in the middle steel truss arch was studied, as shown in Figure 6.

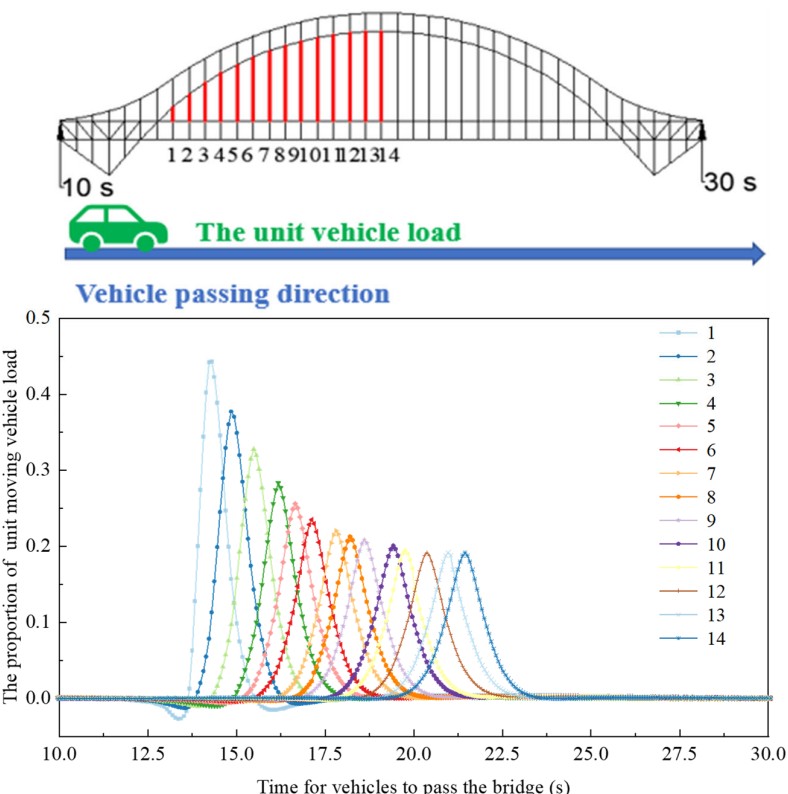

**Figure 6.** The suspender stress amplitude in middle steel truss arch.

It can be seen from Figure 6 that for the half arch suspenders from No. 1 to No. 14, the tensile stress amplitude decreased gradually when the vehicle passed on the bridge. The No. 1 suspender at the arch foot position achieved the largest stress amplitude.

### 5.1. Impact of Steel Truss Arch on the Short Suspender

There are three steel truss arches in the Mingzhu Bay bridge. The stress state of the short suspender in the middle and the side steel truss arch was investigated in the FE model. A number of stress cycles occur in the bridge members as the vehicle moves on the bridge. The dynamic responses of seven types of moving vehicles were considered and the stress amplitude of the suspender was obtained from the FE model. In addition, when the vehicle passes through the main span at a speed of 80 km/h, the time starts from 10 s and ends at 30 s. The stress of the suspender is shown in Figure 7 as the M1–M7 vehicle models passed the main span.

The Miner's cumulative fatigue damage theory indicates that all stress cycle amplitudes and their corresponding times of action need to be obtained for the fatigue assessment of the suspender. Therefore, it is necessary to process the fatigue stress spectrum of the suspender, extract all the cyclic amplitudes, and count the numbers of each cycle. This paper takes the short suspender of the steel truss arch bridge as the research object to evaluate the fatigue performance and predict the fatigue life. In terms of the rain-flow

counting method [34,35], the stress amplitude of the short suspender of the middle truss and the side truss was counted for each cycle and accumulated by MATLAB [36] to generate the stress spectrum, as shown in Figure 8.

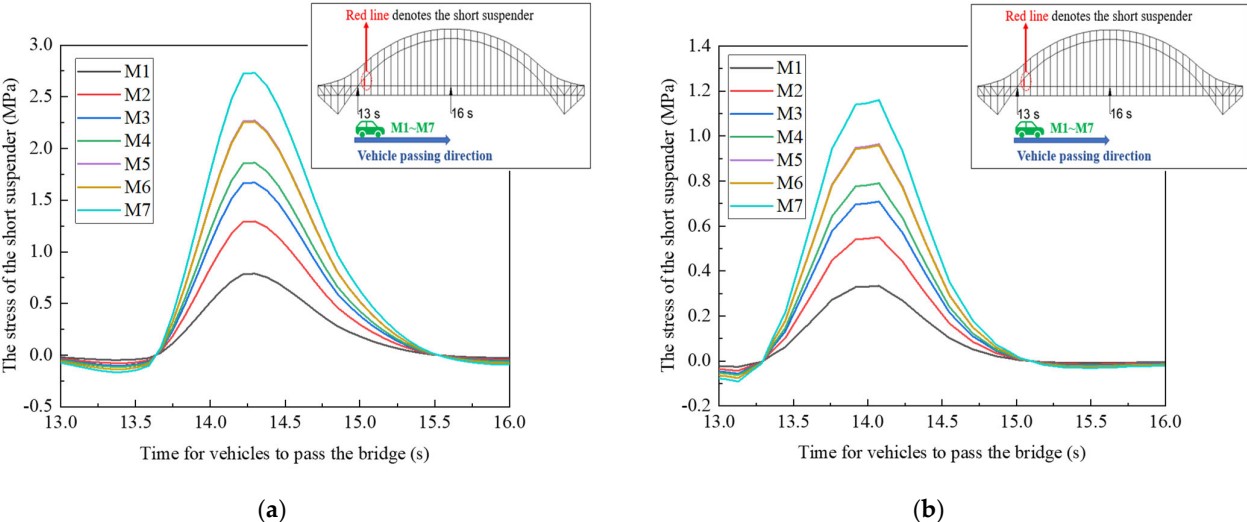

(**a**) (**b**)

**Figure 7.** Dynamic response of the short suspenders under single vehicle load. (**a**) The middle steel truss arch; (**b**) the side steel truss arch.

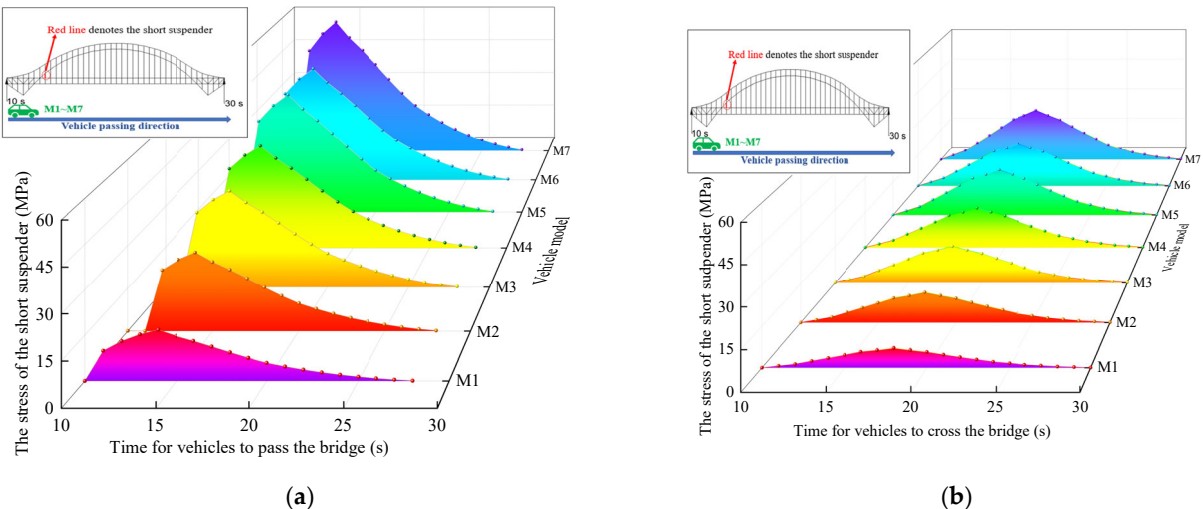

(**a**) (**b**)

**Figure 8.** Dynamic response of the short suspenders under traffic load. (**a**) The middle steel truss arch; (**b**) the side steel truss arch; M1–M7 are vehicle model types.

For the middle steel truss arch, the stress on the suspender is about 50% higher than the suspender in the side steel truss arch with the same vehicle loading. Under the moving vehicle of M7, the maximum stress on the middle and the side suspender is 46.7 MPa and 23.6 MPa, respectively. Moreover, the stress amplitude and the number of cycles were extracted and the data are shown in Table 4.

The fatigue damage of the suspender can be obtained from the *S-N* curve Equation (1) to evaluate the fatigue life of the suspender. From the stress spectrum in Figure 8, the average stress of the short suspender of the middle truss is 274.16 MPa, the ultimate tensile stress of the suspender steel wire is 1670 MPa [37], and the *S-N* curve Equation (1) can be simplified into:

$$\lg N = 14.82 - 3.5 \lg \sigma_a \tag{2}$$

**Table 4.** Stress amplitude statistics of the short suspender per day.

| Stress Amplitude $\sigma_a$ (MPa) | Number of Cycles (Times) | Stress Amplitude $\sigma_a$ (MPa) | Number of Cycles (Times) |
|---|---|---|---|
| 0~1 | 3900 | 25~30 | 0 |
| 1~5 | 1839 | 30~35 | 0 |
| 5~10 | 2061 | 35~40 | 0 |
| 10~15 | 0 | 40~45 | 0 |
| 15~20 | 0 | 45~50 | 0 |
| 20~25 | 0 | >50 | 3900 |

The fatigue damage degree *d* under a single stress cycle can be deduced as

$$\mathrm{d} = \frac{1}{\mathrm{N}} = \frac{\sigma_a^{3.5}}{14.82^{10}} \tag{3}$$

With Miner's linear cumulative fatigue damage theory, the total fatigue damage degree D can be summed by the damage degree d

$$\mathrm{D} = \sum d = \sum \frac{1}{N} = \sum \frac{\sigma_a^{3.5}}{14.82^{10}} \tag{4}$$

$$\sigma_{ae} = \left( \frac{\sum_{i=1}^{x} \sigma_{ai}^m}{x} \right)^{\frac{1}{m}} = \left( \frac{\sum \sigma_a^{3.5}}{n} \right)^{\frac{1}{3.5}} \tag{5}$$

where *n* is the total number of cyclic stresses; $\sigma_{ae}$ is the equivalent stress amplitude of the suspender. Based on the data in Table 4 and Figure 8 the fatigue life of the short suspender can be calculated. Moreover, the fatigue life of the short suspender is 140 years, which meets the design life of 100 years.

### 5.2. Impact of Overloaded Rate

From the stress analysis of the suspenders from the middle truss and the side truss, it was found that the stress on the suspender in the middle truss arch was always larger than the suspenders in the side truss arch. Hence, the fatigue life assessment was conducted on the middle truss arch suspenders. The overload moving vehicles rate was investigated by a standard, 25%, and 50% overload with seven fatigue vehicle types; the overweight data and traffic flow are shown in Table 5.

**Table 5.** Overloaded vehicle data.

| Vehicle Model | Standard | Overweight Rate | | Traffic Flow Per Day |
|---|---|---|---|---|
| | Weight (kN) | 25% (kN) | 50% (kN) | |
| M1 | 135 | 168.75 | 202.5 | 1543 |
| M2 | 222 | 277.5 | 333 | 223 |
| M3 | 286 | 357.5 | 429 | 73 |
| M4 | 319 | 398.5 | 478.5 | 460 |
| M5 | 389 | 486.25 | 583.5 | 13 |
| M6 | 387 | 483.75 | 580.5 | 135 |
| M7 | 486 | 585 | 729 | 1453 |

The fatigue life of the suspenders on the half-span arch were evaluated under moving vehicle standard, 25%, and 50% overloaded rates (Figure 9).

It can be seen from Figure 9 the fatigue life of the suspender can meet the design life of 100 years under the standard fatigue load. With the overload rate up to 50%, the fatigue life of the suspender is lower than the design service life. For the short suspender along the bridge direction, the suspender decreases significantly, and its amplitude is about 50 years, which cannot meet the requirement of the design. The fatigue life of the suspender shows a non-linear decreasing trend with the increasing overloading ratio. Furthermore, the fatigue

life of the suspender decreases by about 30%, which shows that the overload condition has a significant impact on the fatigue life of the suspender in the steel truss arch bridge.

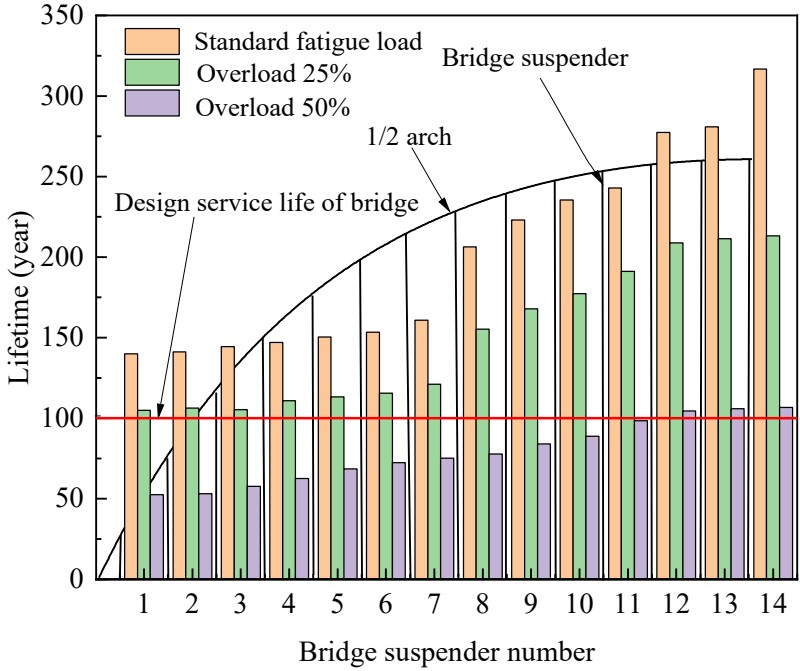

**Figure 9.** Fatigue life comparison of the suspenders in middle truss.

## 6. Conclusions

Based on the Guangzhou Mingzhu Bay bridge, the whole bridge model was established by the finite element software Midas Civil. The fatigue life of the suspender was evaluated under a standard fatigue load, and 25% and 50% overloading; the conclusions were drawn as follows:

1. The short suspenders near the arch foot show a larger stress amplitude than others, while the stress on the suspenders in the middle steel truss arch is greater than the suspenders in the side truss arch. However, the fatigue life of the short suspender under a standard traffic load is 140 years, which meets the design requirement of 100 years.
2. The fatigue life of the suspenders decreases by 20% and 30% under the overloaded rate of 25% and 50%, respectively. With a vehicle design speed of 80 km/h, the fatigue life of the short suspender on the mid-truss arch is about 50 years under the 50% overloaded rate, which is significantly lower than the bridge design life of 100 years.
3. The fatigue life shows a non-linear decreasing trend with the increasing overloading ratio. Overloading has a significant effect on the suspender life of steel truss arch bridges. Herein, the overloaded vehicles should be strictly controlled to ensure the safety of the bridge during its service.

The conclusions are based on the FE model analysis and Miner's cumulative fatigue damage theory. If we suppose that, at the specified stress level, the accumulation of fatigue damage is independent of the previous loading history of the material, then the damage from a single loading cycle should be the same, which means the fatigue life is not affected by the loading sequence regardless of whether the stress amplitude is from high to low or from low to high. However, the actual situation is that the loading sequence has a certain impact on the fatigue life of the components. Because the fatigue failure mechanism is under the same stress level, the damage degree with the microcrack formation and main crack growth is different. Therefore, the theoretical analysis results are usually not all the same as the test data in the construction.

**Author Contributions:** P.L., H.L. and Y.C. wrote the manuscript together; J.Z. and L.A. conducted data processing and drawings; Y.W. and J.L. revised the manuscript. All authors have read and agreed to the published version of the manuscript.

**Funding:** Science and Technology Research and Development Project of China Railway Construction Bridge Engineering Bureau Group Co., LTD. (DQJ-2018-A01), Tianjin Science and Technology Development Plan Project (19YDLZSF00030), National Natural Science Foundation of China (51038006).

**Institutional Review Board Statement:** Not applicable.

**Informed Consent Statement:** Not applicable.

**Data Availability Statement:** Not applicable.

**Acknowledgments:** The authors would like to thank China Railway Construction and Bridge Bureau Group Co., LTD. for the experiment support.

**Conflicts of Interest:** The authors declare no conflict of interest.

## Nomenclature

| | |
|---|---|
| $N$ | number of fatigue cycles, |
| $\sigma_m$ | mean stress value of $\sigma_a$ and $\sigma_b$, |
| $\sigma_a, \sigma_b$ | tensile strength of the suspenders and the mean stress of the suspenders, |
| $d, D$ | fatigue damage per single cycle and fatigue damage per day, |
| $m$ | coefficient of the fatigue damage, |
| $x$ | number of cycles to reach a stress level, |
| $n$ | number of vehicle cycles, |
| $\sigma_{ae}$ | the equivalent stress amplitude of the suspender. |

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
