# Peer review of "Fatigue Analysis of Long-Span Steel Truss Arched Bridge Part II: Fatigue Life Assessment of Suspenders Subjected to Dynamic Overloaded Moving Vehicles"

_metals, doi:10.3390/met12061035_

Round 1
Reviewer 1 Report
The present manuscript studied fatigue life sssessment of suspenders on long-span steel arched bridge. I recommend the paper for publication after minor improvements. The paper reports an interesting and very useful work, well structured in the manuscript, but the manuscript has some weaknesses. Mentioned below aspects must be taken into consideration during the revision:
Nomenclature:
(1) I suggest adding "Nomenclature" section (with units and abbreviations) in the manuscript.
Introduction:
(2) Literature analysis should be expanded. A lot of works dealing with this issue have been published (with an emphasis on the fatigue life assessment models), especially in Metals journal. Please see point (8) for the suggested papers.
Materials and Methods:
(3) Could You include the chemical composition of investigated steel (from literature or determined by the authors for the analysed material);
(4) Please provide information source about mechanical parameters from literature (if any);
(5) It would be advisable to show the whole method algorithm in a flowchart.
Results:
(6) The main limitations of the present method must be identified and discussed in the end of this section.
(7) Are the fatigue life results similar to those obtained by other authors in the literature (if any)?
References:
(8) References section should be extended. I propose to add a few entries in the Introduction section regarding the fatigue life analysis (Carpinteri et al., 2016; Macek et al., 2017; Pejkowski and Seyda, 2021)
- Carpinteri, A., Fortese, G., Ronchei, C., Scorza, D., Spagnoli, A., Vantadori, S., 2016. Fatigue life evaluation of metallic structures under multiaxial random loading. International Journal of Fatigue 90, 191–199. https://doi.org/10.1016/j.ijfatigue.2016.05.007
- Macek, W., Łagoda, T., Mucha, N., 2017. Energy-based fatigue failure characteristics of materials under random bending loading in elastic-plastic range. Fatigue and Fracture of Engineering Materials and Structures. https://doi.org/10.1111/ffe.12677
- Pejkowski, Ł., Seyda, J., 2021. Fatigue of four metallic materials under asynchronous loadings: Small cracks observation and fatigue life prediction. International Journal of Fatigue 142, 105904. https://doi.org/10.1016/J.IJFATIGUE.2020.105904
Author Response
"Please see the attachment."

Reviewer 2 Report
Dear authors,
enclosed you find my comments on your paper entitled “Fatigue Life Assessment of Suspenders on Long-Span Steel Arched Bridge Subjected to Dynamic Overloaded Moving Vehicles”.
The text should be basically revised, there are some typos and different spellings for some words e.g. half-through. Overall, the text is difficult to read. Please always put a space between number and unit.
Figure labels should be consistent. For example, in Fig. 4 there are three different ways of starting the labelling for a), b) and c). In addition, the illustrations should also be labelled uniformly, e.g. generally with capital letters at the beginning.
Table 1 is difficult to read, please label it larger.
In general, figures or tables should always be on the same page (e.g. Fig. 4 and Table 1).
Figure 5 is distorted and labelled too small.
Figure 6 and 7 are difficult to read, especially the axes.
Equations 3-1 to 3-6 are presented in a very unclear way. The authors should also provide more information on what is calculated here as individual terms. In addition, shouldn't the numbering start with 4 in the section?
What is shown on the y-axis in Fig. 8? The labelling is missing here.
Chapter 5.3 lacks context for the rest of the text. How does one arrive at this result based on the previous sections? Please show how the individual steps are connected.
A discussion of the results is missing in the paper. This should be integrated into the results or added as a separate section.
The summary only refers to a small section of the paper. More details should be listed here.
Author Response
"Please see the attachment."

Round 2
Reviewer 2 Report
Dear authors,
Figures 1, 6, 7 and 8 are still barely readable.
There is a gap on page 5.
Table 4 is spread over 2 pages (please also adjust: In the tables headline the bracket is open in one line and MPa) in the next, is not wrong but unsightly).
In 4.2 an assignment to the reference is missing [Error!...].
In 5. an assignment to the reference is missing [Error!..].
5.1 lacks an assignment to the reference [Error!...].
And a lot more of them (7 times!!)
In line 120, a space is missing before [28,29].
Line 222 is missing a space before [39].
The spacing in line 121 is not clear
The spacing in line 223/224 is not clear.
The spacing in line 209 is not clear.
Line 226 Simplified with lower case s
Line 216 MPa instead of MP
Line 216/217, spaces are missing before units
Punctuation marks are partly missing
The text should be fundamentally revised.
In my view, the revision is not really successful. For this reason, I cannot approve the present version for publication.
